# On-Board Decentralized Observation Planning for LEO Satellite Constellations

**Bingyu Song \*, Yingwu Chen, Qing Yang, Yahui Zuo, Shilong Xu and Yuning Chen**

College of Information System and Management, National University of Defense Technology, Changsha 410073, China
\* Correspondence: sbynudt@nudt.edu.cn

**Abstract:** The multi-satellite on-board observation planning (MSOOP) is a variant of the multi-agent task allocation problem (MATAP). MSOOP is used to complete the observation task allocation in a fully cooperative mode to maximize the profits of the whole system. In this paper, MSOOP for LEO satellite constellations is investigated, and the decentralized algorithm is exploited for solving it. The problem description of MSOOP for LEO satellite constellations is detailed. The coupled constraints make MSOOP more complex than other task allocation problems. The improved Consensus-Based Bundle Algorithm (ICBBA), which includes a bundle construction phase and consensus check phase, is proposed. A constraint check and a mask recovery are introduced into bundle construction and consensus check to handle the coupled constraints. The fitness function is adjusted to adapt to the characteristics of different scenes. Experimental results on series instances demonstrate the effectiveness of the proposed algorithm.

**Keywords:** multi-satellite on-board observation planning; task allocation; decentralized; improved consensus-based bundle algorithm

## 1. Introduction

With the increase in the number of satellites and the improvement of satellite capabilities, users have more complex requirements for Earth Observation Satellites (EOSs), and the drawbacks of the traditional observation planning mode, which is conducted entirely on the ground, are increasingly exposed. In the traditional mode, due to the limited communication resources between the satellites and the ground stations, the ground cannot obtain the on-board status in real time, which makes it difficult to perform a practical planning scheme [1]. The improvement of satellite computing power and inter-satellite communication capacity makes on-board planning possible. The advantages of on-board planning are that the cost of satellite–ground interaction can be reduced, and more importantly, more effective planning schemes can be made based on the real-time environmental conditions.

Due to the limited storage, energy and field of view of single satellite, multi-satellite coordination is required to complete complex observation requirements. The satellite constellation is constituted by multiple satellites, which cooperate with each other in accomplishing the specific missions. LEO satellite constellations are widely used in remote sensing (MicroMAS [2], TROPICS [3]) and communication (Iridium [4], Globalstar [5], OneWeb [6], Starlink [7]). Compared with other types of constellation, LEO satellite constellations have several main advantages for remote sensing: (1) LEO satellites are usually small satellites and their lower cost to develop and launch enables organizations to field many dedicated nodes on-orbit; (2) due to the large number of satellites, the revisit period of the constellation to the ground target is shorter; (3) the low orbit altitude makes the delay and loss of Satellite–Ground Link (SGL) lower; (4) in addition to SGL, Inter-Satellite Link (ISL) is basically used in LEO satellite constellations, which allows for crosslink with higher quality and larger bandwidth. In this paper, we investigate on-board observation planning for crosslink-enabled LEO satellite constellations.

On-board observation planning for LEO satellite constellations belongs to the multi-satellite on-board observation planning (MSOOP) which can be seen as a variant of the multi-agent task allocation problem (MATAP) [8]. In MATAP, there are two typical planning architectures—centralized [9–11] and decentralized [12–15]. In a centralized architecture, a central node is required to maximize or minimize user-defined goals by optimizing task-resource allocation, expecting to find the optimal solution. The centralized optimization is built on the premises that the central node knows all the task information and the resource information of the global nodes. This architecture cannot respond to changes on the satellites in real time, which leads to a certain lag in the planning scheme, and it relies heavily on communication links. In addition, the computing power of on-board hardware is poor, so it is difficult to find a node to complete the planning calculation of the whole constellation, especially a large-scale constellation. Thus, it is more feasible to allocate the calculation cost to multiple nodes. Finally, considering the robustness of the whole system, once the central node fails, it will be a devastating blow to the entire constellation. To sum up, a centralized architecture is hard to implement in MSOOP, and a decentralized architecture is chosen. In a decentralized architecture, each satellite claims the task independently and finally generates its own planning scheme. Following that, the consensus problem is difficult to deal with. Assuming that there is no communication between satellites, it cannot be guaranteed that a task will not be claimed repeatedly by multiple satellites, because a satellite does not know the claim results of other nodes. Therefore, we must introduce communication between satellites to ensure consensus. In general, if there is communication so that each satellite knows the claim results of other satellites, consensus can be guaranteed. The communication in LEO satellite constellations is achieved by ISLs. However, as a limited resource, the link cannot be used at will, and there is latency and packet loss in the communication process, which can be seen as the cost of communication.

In previous literature, traditional decentralized optimization algorithms mainly include meta-heuristic algorithms and market-based heuristic algorithms. For meta-heuristic algorithms, Zheng et al. developed a hybrid distributed Genetic Algorithm including a local constraint satisfaction module and a global distributed optimization module [1]. The local module is performed on each satellite using a local search to provide feasible sub-solutions, and the global module is performed through the inter-satellite communication using a distributed Genetic Algorithm to ensure the satisfaction of coupled global constraints. However, the computational complexity for both the local module and the global module grows exponentially as the number of satellites increases. Agogino et al. exploited a multi-agent evolutionary algorithm to deal with the coordination of CubeSats [16]. Similarly, the complexity of evolutionary computation is uncontrollable in large-scale scenes, and some time constraints are not considered. To sum up, meta-heuristic algorithms usually include some iterative optimization mechanisms, such as local search and population evolution, which makes the computational complexity rise nonlinearly as the number of agents or tasks increases and leads to a poor performance in large-scale scenes. In addition, the computing power of on-board hardware is not enough to support iterative optimization mechanisms. For market-based heuristic algorithms, Van der Horst studied how to optimally manage interdependent satellite clusters under the constraints of energy and communication for heterogeneous small satellite systems, and they introduced a task allocation method based on market mechanism [17]. However, the model is relatively simple, and some coupled constraints such as temporary constraints are not considered. As a representative market-based algorithm, Contract Network Protocol (CNP) was first proposed to deal with distributed problem by Smith [18]. After that, many improved versions have been studied for satellite mission planning [19–22]. However, the communication cost of CNP and its improved versions cannot be well controlled within an acceptable level. Each task needs to go through the stages of publishing, bidding and winning, which makes the number of communication for task allocation linearly related to the number of agents and the number of tasks. For the whole constellation, a large amount of communication makes the communication cost

increase and the robustness decrease. In addition to traditional optimization algorithms, some machine learning algorithms are also emerging to solve decentralized problems. These algorithms are usually based on Game Theory, which achieves global convergence by finding the Nash equilibrium. Multi-Agent Reinforcement Learning (MARL) is widely used in multi-agent systems, such as games (StarCraft II), robot control, and UAV formation. Li et al. introduced Multi-Agent Deep Deterministic Policy Gradient (MADDPG), a classical MARL algorithm, into MSOOP which was formulated as a fully cooperative Markov Decision Process [23]. However, only the coordination of three satellites is considered in their experiment. It is difficult to guarantee the convergence of machine learning algorithms in large-scale scenes. Therefore, considering the computational complexity, communication cost and convergence, we decide to explore a market-based decentralized algorithm with a low communication cost.

In this paper, the Improved Consensus-Based Bundle Algorithm (ICBBA) is proposed to maximize the total profit of the whole constellation at an acceptable communication cost. CBBA is one of the state-of-the-art decentralized market-based algorithms that was developed by Choi et al. in 2009 [24]. CBBA is a polynomial time algorithm with $O(N \cdot M)$ computational complexity ($N$ is the number of agents and $M$ is the number of tasks). It has been proved that CBBA is capable of guaranteeing conflict-free allocation, assuming a strongly connected network, which is one type of coupled constraint. However, it is not able to account for the other types of coupling which are common in MSOOP. ICBBA is an innovation based on CBBA in MSOOP, which is reflected in the following aspects: (1) aiming at other coupled constraints such as the storage constraint, the energy constraint and the temporary constraints in MSOOP, a constraint check and a mask recovery are introduced into bundle construction and consensus check to overcome the shortcoming that the original algorithm cannot solve the coupled constraints; (2) according to the characteristics of different scenes, the fitness function is adjusted to adapt different dominant constraints. To validate the effectiveness of ICBBA, we carry out experiments on several groups of instances with different satellite and task scales. The experimental results show that ICBBA can not only reduce the communication cost compared with other decentralized algorithms but also have a good optimization performance. ICBBA can achieve the global consensus in an acceptable number of iterations and the re-convergence in the face of dynamic task arrival when it is running.

The remainder of the paper is organized as follows. Section 2 gives the description of MSOOP. ICBBA including bundle construction and consensus check is introduced in Section 3. Section 4 presents the design of test instances and the experimental results of ICBBA on various scenes. Concluding remarks are drawn in Section 5.

## 2. Problem Description

In this section, the problem description of MSOOP for LEO satellite constellations is detailed. The additional discussion on the design of satellite constellations will not be conducted in this paper. The homogeneous Walker-$\delta$ constellation consisting of the same optical satellites is used as the studied case. The constellation configuration is described in Section 2.1; the mathematical formulation of MSOOP is described in Section 2.2; the constellation communication network topology is described in Section 2.3.

### 2.1. Constellation Configuration

A Walker-$\delta$ constellation configuration can be represented as the parameter tuple (total number of satellites $N$/number of planes $P$/inter-plane spacing $F$, orbit altitude $h$, orbit inclination $i$). The total number of satellites, the number of planes, and the inter plane spacing determine the scale of the whole constellation. Each satellite in the constellation has the same orbit altitude and orbit inclination. In this paper, AGI's STK is introduced to simulate all the constellations. Taking Walker-$\delta$ (30/3/1, 600 km, 60°) as an example, its 3D constellation configuration and 2D satellite ground tracks at time $t = 0$ are shown in Figure 1a,b respectively.

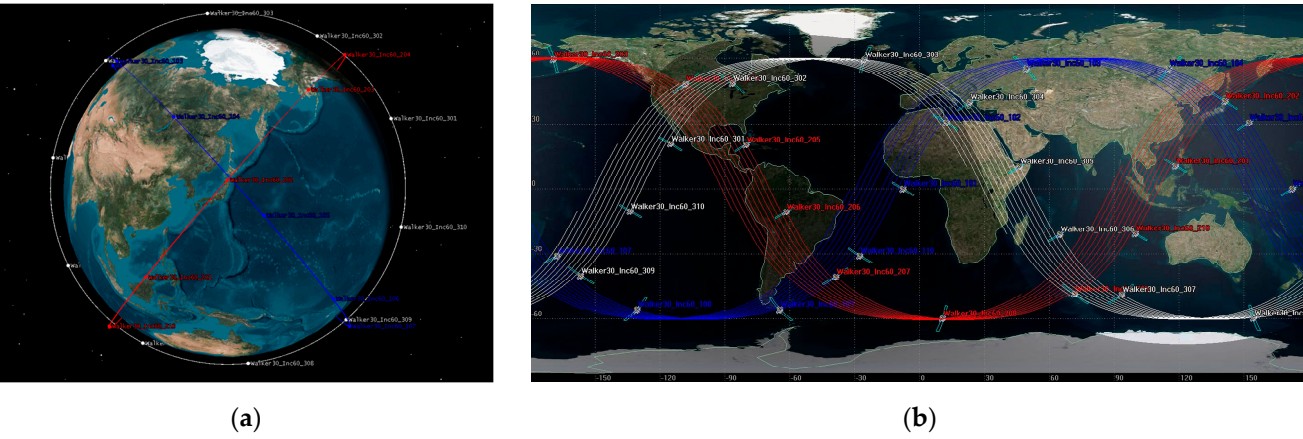

**Figure 1.** Walker-δ (30/3/1, 600 km, 60°) constellation configuration: (**a**) 3D constellation configuration at time $t = 0$; (**b**) 2D satellite ground tracks at time $t = 0$.

For each satellite in the constellation, it has the same attitude maneuver capability and carries the same observation and communication payload. The observation payload configuration includes its main parameters such as payload type, field of view and resolution. The satellite can only observe the target within the corresponding observation availability masks. The satellite-to-target observation availability masks are jointly determined by attitude maneuver capability, observation payload configuration, satellite position and target position. Communication payloads such as radio or laser payloads enable communication links between a satellite and a ground station or between two satellites to transmit data. However, since the relative positions of any two nodes in the communication network may change over time, they must be within the communication availability masks to establish communication links.

### 2.2. Mathematical Formulation

In MSOOP, the constellation handles demands from ground users or generated spontaneously, and these demands can be considered as the set of observation tasks. Generally speaking, the set of observation tasks and the set of observation availability masks are taken as the input, and the planning scheme is taken as the output. According to the on-board situation, each satellite completes the task allocation in a fully cooperative mode, which maximizes the profits of the whole constellation to complete the observation tasks under the condition of meeting various constraints. Therefore, MSOOP is essentially a variant of MATAP, so we will modify and supplement the task allocation model in the original work of CBBA to formulate the mathematical model of MSOOP.

- Satellite-related variables: Let $\mathcal{I} = \{1, 2, \ldots, N\}$ be index set of $N$ satellites. The $i$th satellite $sat_i$ is associated with the maximum available storage $sto_i^{\text{sat}}$ and the maximum available energy $ene_i^{\text{sat}}$.

- Task-related variables: Let $\mathcal{J} = \{1, 2, \ldots, M\}$ be an index set of $M$ tasks. The $j$th task $task_j$ is associated with the estimated consumed storage $sto_j^{\text{task}}$, the estimated consumed energy $ene_j^{\text{task}}$, the estimated duration $dur_j^{\text{task}}$, the execution start time $st_j^{\text{task}}$, the execution end time $et_j^{\text{task}}$, the attitude maneuver time $amt_{jj'}^{\text{task}}$ between $et_j^{\text{task}}$ and $st_{j'}^{\text{task}}$, the original profit $op_j^{\text{task}}$ determined by the priority of the task, and the final profit $fp_j^{\text{task}}$ obtained by the observation of the task. $\lambda$ is the time-discounted parameter of $fp_j^{\text{task}}$, and $fp_j^{\text{task}}$ can be calculated as:

$$fp_j^{\text{task}} = op_j^{\text{task}}e^{-\lambda st_j^{\text{task}}}. \tag{1}$$

- Mask-related variables: Let $\mathcal{K}_{ij} = \{1, 2, \ldots, O_{ij}\}$ be index set of $O_{ij}$ $sat_i$-to-$task_j$ observation availability masks. The $k$th mask $oam_{ijk}$ is associated with the start time $st_{ijk}^{\text{oam}}$ and the end time $et_{ijk}^{\text{oam}}$.
- Decision variable: To finally find the "task-mask" mappings in MSOOP, $x_{ijk}$ is introduced as a decision variable. $x_{ijk} = 1$ if $task_j$ is assigned to $oam_{ijk}$, and $x_{ijk} = 0$ otherwise.

Based on the above variable definitions, the mathematical model of MSOOP is formulated as follows:

$$\max \sum_{i=1}^{N} \left( \sum_{j=1}^{M} \left( \sum_{k=1}^{O_{ij}} x_{ijk} f p_j^{\text{task}} \right) \right) \tag{2}$$

subject to

$$\sum_{j=1}^{M} \left( \sum_{k=1}^{O_{ij}} x_{ijk} sto_j^{\text{task}} \right) \leq sto_i^{\text{sat}}, \ \forall i \in \mathcal{I} \tag{3}$$

$$\sum_{j=1}^{M} \left( \sum_{k=1}^{O_{ij}} x_{ijk} ene_j^{\text{task}} \right) \leq ene_i^{\text{sat}}, \ \forall i \in \mathcal{I} \tag{4}$$

$$\sum_{i=1}^{N} \left( \sum_{k=1}^{O_{ij}} x_{ijk} \right) \leq 1, \ \forall j \in \mathcal{J} \tag{5}$$

$$st_{ijk}^{\text{oam}} \leq st_j^{\text{task}} < et_j^{\text{task}} \leq et_{ijk}^{\text{oam}}, \ \text{if } x_{ijk} = 1 \tag{6}$$

$$et_j^{\text{task}} + amt_{jj'}^{\text{task}} \leq st_{j'}^{\text{task}}, \ \text{if } et_j^{\text{task}} < st_{j'}^{\text{task}} \tag{7}$$

$$st_j^{\text{task}} + dur_j^{\text{task}} = et_j^{\text{task}}, \ \text{if } \sum_{i=1}^{N} \left( \sum_{k=1}^{O_{ij}} x_{ijk} \right) = 1 \tag{8}$$

$$x_{ijk} \in \{0, 1\}, \ \forall (i, j, k) \in \mathcal{I} \times \mathcal{J} \times \mathcal{K}_{ij} \tag{9}$$

where Formula (2) represents that the objective function is to maximize the total profit; Formulas (3) and (4), respectively, use the storage constraint and the energy constraint to restrict the number of tasks completed by each satellite; Formula (5) describes the consensus constraint; Formulas (6)–(8) describe the special temporary constraints in MSOOP, Formula (6) means the observation action of each planned task must be within the observation availability mask; Formula (7) means any two observation actions do not overlap and the attitude maneuver time between them should be reserved; Formula (8) means there must be a duration needed for each planned task to receive a complete observation action; Formula (9) guarantees that once an observation action starts, it cannot be preempted.

According to attitude maneuver capability, Earth Observation Satellites are divided into non-agile satellites and agile satellites. Non-agile satellites have one axe of the single direction (roll), which means the target can only be imaged when the satellite is just on the top of target point. In Formula (6), if $task_j$ is assigned to $oam_{ijk}$, the execution start time $st_j^{\text{task}}$ should be equal to $st_{ijk}^{\text{oam}}$ and the execution end time $et_j^{\text{task}}$ should be equal to $et_{ijk}^{\text{oam}}$. Agile satellites are mobile along three axes (roll, pitch and yaw). This mobility makes the target also can be imaged even if the satellite flies over or does not reach the target point. Agile satellites have the longer observation availability mask for a task than non-agile satellites. So, we need to determine not only $x_{ijk}$ but also $st_j^{\text{task}}$ and $et_j^{\text{task}}$ of each task. Observation planning for single agile satellite has been proved to be NP-hard [25]. $st_j^{\text{task}}$ and $et_j^{\text{task}}$ are continuous variables, so it is difficult to use discrete algorithms to determine. In this paper, we use heuristic strategies such as forward arrangement and central arrangement to determine the execution time of each task.

The above constraints are divided into coupled constraints and uncoupled constraints. Coupled constraints include any situation where the decisions regarding one task or agent

affect the options available regarding other tasks or agents. The constraints described in Formulas (3)–(5) and (7) are coupled constraints. The observation action of each task may cause other observation masks to be disabled due to insufficient storage or energy, avoidance of re-assignment, or lack of attitude maneuver time. These coupled constraints make MSOOP more complex than other task allocation problems.

*2.3. Communication Network Topology*

In MSOOP, the communication links are used to carry out data routing related to the planning process, such as broadcast of task sets, interaction of consensus information, etc. The size of these data is usually small, which is different from bulk data such as images. Therefore, there is no strict demand for the bandwidth of the communication links. The availability of the communication links is mainly discussed here. For any two nodes at a certain time, if a communication link can be established between them, they are defined to be neighbors to each other, and this relationship is bi-directional.

- Communication-related variables: Let $\mathcal{L}_{ii'} = \{1, 2, \ldots, P_{ii'}\}$ be an index set of $P_{ii'}$ $sat_i$-to-$sat_{i'}$ communication availability masks. The $l$th mask $cam_{ii'l}$ is associated with the start time $st_{ii'l}^{cam}$ and the end time $et_{ii'l}^{cam}$. If $\exists l \in \mathcal{L}_{ii'}$ satisfies $st_{ii'l}^{cam} \leq t \leq et_{ii'l}^{cam}$ at time $t$, $sat_{i'}$ is called the neighbor of $sat_i$ at time $t$. The index set of the neighbors of $sat_i$ at time $t$ is denoted as $\mathcal{N}_i^t = \{1, 2, \ldots, Q_i^t\}$, where $Q_i^t$ is the number of the neighbors of $sat_i$ at time $t$. Due to the bi-directional characteristic, $\mathcal{L}_{ii'} = \mathcal{L}_{i'i}$ and $i' \in \mathcal{N}_i^t \equiv i \in \mathcal{N}_{i'}^t$.

The crosslink between satellites is realized by establishing ISLs. ISLs are divided into intra-plane ISLs and inter-plane ISLs according to whether the sender and the receiver belong to the same plane. Each satellite can establish a stable intra-plane ISL with satellites on both sides of the same plane at any time, which makes the intra-plane ISLs form a closed loop. The availability of inter-plane ISLs is time-varying due to the relative motion of satellites on different planes. In this paper, we make a reasonable simplified availability standard that an ISL is available whenever the line-of-sight vector between two satellites passes above the surface of the earth. The communication network topology of Walker-δ (30/3/1, 600 km, 60°) at time $t = 0$ is shown in Figure 2. For instance, $sat_1$ has two intra-plane neighbors ($sat_2$, $sat_{10}$). $sat_2$ has two intra-plane neighbors ($sat_1$, $sat_3$) and three inter-plane neighbors ($sat_{23}$, $sat_{24}$, $sat_{25}$).

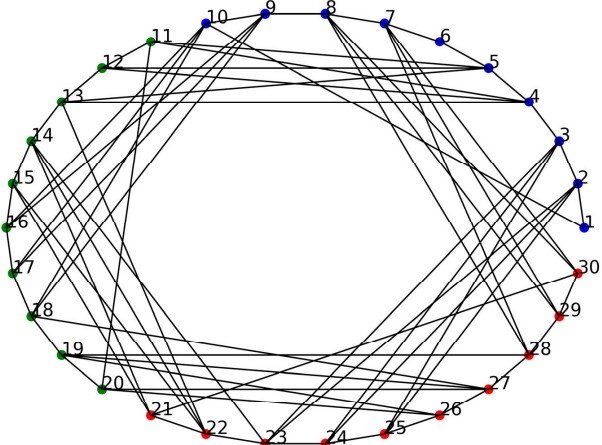

**Figure 2.** The communication network topology of Walker-δ (30/3/1, 600 km, 60°) at time $t = 0$ (connect any two satellites between which ISL is available).

## 3. Improved CBBA

In order to balance the computing cost of nodes and increase the overall robustness, we decided to adopt a decentralized solving approach. In this section, ICBBA which is modified based on the original CBBA is proposed to solve MSOOP. The main procedure of

ICBBA is described in Section 3.1; the bundle construction is described in Section 3.2; and the consensus check is described in Section 3.3.

### 3.1. Main Procedure of ICBBA

Before implementing the ICBBA, some pre-work is essential. The ground needs to carry out periodic orbit prediction and communication availability calculation in advance and send it to the constellation before planning. No matter whether the observation tasks are generated on the ground or on the satellite, they need to be broadcast in the constellation in advance. Each satellite calculates the observation availability masks based on satellite information and task information before planning. After all satellites obtain the observation availability masks and the communication availability masks mentioned in Sections 2.2 and 2.3, respectively, ICBBA can be started.

ICBBA is a typical market-based task allocation algorithm, which includes two main phases. The first phase is bundle construction, in which all nodes independently add tasks greedily to their bundle. The second phase is consensus check, in which nodes communicate with their neighbors to reduce conflict in the global task allocation based on action rules. On the premise of keeping each satellite clock in sync with the ground, each satellite runs the above two phases iteratively in parallel until the global consensus is reached and receives its own final planning scheme to perform observation. For the decentralized approach, each node cannot perceive the global planning information, and it can only receive the planning information from its neighbors. Therefore, each satellite actually does not know whether the global consensus has been reached (that is, each satellite does not know whether the global algorithm has been terminated) but only whether the consensus between itself and its neighbors has been reached. Although there is no centralized node for unified planning, as long as each decentralized node operates correctly, the global convergence can still be achieved. The convergence of CBBA has been proved in the original work [24]: for dynamic networks in which the communication network topology varies with time, the convergence of CBBA with a conflict resolution phase can still be guaranteed if the union of network topologies for the effective communication of each agent is fully connected. As we introduced in Section 2.3, intra-plane ISLs and inter-plane ISLs allow each satellite node to be fully connected within ICBBA running time, and no node is isolated.

- Algorithm-related variables: $T$ is the iteration time that records the running times of bundle construction. $\mathbf{b}_i$ is the bundle of $sat_i$ with the length of $B_i$, where the element $b_{in}$ is the $n$th entry of $\mathbf{b}_i$. $\mathbf{y}_i$ is the winning bid list of $sat_i$, where the element $y_{ij}$ is the global winning bid for $task_j$ with the knowledge of $sat_i$. $\mathbf{z}_i$ is the winning satellite list of $sat_i$, where the element $z_{ij}$ is the global winning satellite index for $task_j$ with the knowledge of $sat_i$. $\mathbf{s}_i$ is the time stamp of the last information update from each of the other satellites, where the element $s_{ii'}$ is the last information-update time stamp of $sat_{i'}$ with the knowledge of $sat_i$. $DMG_{ij}$, Diminishing Margin Gain, indicates that if $task_j$ is assigned to $sat_i$, the maximum $fp_j^{\text{task}}$ that can be obtained without conflict with the tasks is already added to $\mathbf{b}_i$. The mask index and the task execution start time corresponding to $DMG_{ij}$ are denoted as $k_{ij}^{\text{DMG}}$, $st_{ij}^{\text{DMG}}$, respectively.

  Each time, the task index added to $\mathbf{b}_i$ is denoted as $j^{\text{add}}$. With the addition of tasks, some masks belonging to the tasks that have not been added to the bundle may be unavailable due to constraint conflicts. Therefore, a mask enabling flag $f_{ijk}$ initialized to 1 is introduced; $f_{ijk} = 1$ if $oam_{ijk}$ is available, $f_{ijk} = 0$ otherwise. A consensus flag $conf_i$ is also introduced; $conf_i = 1$ if the consensus is reached between $sat_i$ and its neighbors currently, $conf_i = 0$ otherwise. $h_{ij}$ is used as an indicator function that is 1 if $DMG_{ij} > y_{ij}$ is true and 0 otherwise, which is expressed as:

$$h_{ij} = \mathbb{I}(DMG_{ij} > y_{ij}). \tag{10}$$

From the perspective of each decentralized node, each satellite starts to run the ICBBA algorithm after receiving the task set. The main procedure of ICBBA includes the following steps:

(1) Bundle Construction: construct its own bundle and update planning information $\mathbf{b}_i$, $\mathbf{y}_i$, $\mathbf{z}_i$; (2) Planning Information Sending: send its own planning information $\mathbf{y}_i$, $\mathbf{z}_i$, $\mathbf{s}_i$ to neighbors; (3) Planning Information Receiving: receive the planning information $\mathbf{y}_{i'}$, $\mathbf{z}_{i'}$, $\mathbf{s}_{i'}$ from its neighbor $sat_{i'}$; (4) Consensus Check: update the time stamp $\mathbf{s}_i$ and check whether the consensus has been reached, no operation if so; otherwise, change the corresponding part of $\mathbf{b}_i$, $\mathbf{y}_i$, $\mathbf{z}_i$ and jump to Step (1).

Step (1) is triggered by non-consensus. Step (2) is triggered by Step (1). Step (3) is triggered by Step (2) of neighbor $sat_{i'}$. Step (4) is triggered by Step (3). Because each satellite does not know whether the global consensus has been reached, the ICBBA on each satellite cannot be terminated spontaneously until the earliest task execution start time or the running time threshold set in advance is reached. After the algorithm is terminated, each satellite performs the observation according to the last updated decentralized planning scheme. Therefore, sufficient running time must be reserved to ensure that the algorithm has completed global convergence; otherwise, the consensus of the planning scheme cannot be guaranteed. The pseudo-code of ICBBA from the perspective of each decentralized node is described in Algorithm 1.

---

**Algorithm 1.** ICBBA for $sat_i$

---

**procedure** ICBBA (current satellite index $i$, satellite index set $\mathcal{I} = \{1, 2, \ldots, N\}$, task index set $\mathcal{J} = \{1, 2, \ldots, M\}$, observation availability mask index sets $\{\mathcal{K}_{i1}, \mathcal{K}_{i2}, \ldots, \mathcal{K}_{iM}\}$, communication availability mask index sets $\{\mathcal{L}_{i1}, \mathcal{L}_{i2}, \ldots, \mathcal{L}_{iN}\}$)

  Initialize $T = 0$, $\mathbf{b}_i(T) = \varnothing$, $y_{ij}(T) = 0 \; \forall j \in \mathcal{J}$, $z_{ij}(T) = -1 \; \forall j \in \mathcal{J}$, $s_{ii'} = 0 \; \forall i' \in \mathcal{I}$, $conf_i = 0$
  **while** termination condition is not reached **do** /* Execute the procedure if the earliest task execution start time or the running time threshold set in advance has not been reached */
    **if** $conf_i = 0$ **then**
      $T = T + 1$ /* Record the iteration time */
      $\mathbf{b}_i(T)$, $\mathbf{y}_i(T)$, $\mathbf{z}_i(T) \leftarrow$ Bundle Construction($i$, $\mathcal{J}$, $\{\mathcal{K}_{i1}, \mathcal{K}_{i2}, \ldots, \mathcal{K}_{iM}\}$, $\mathbf{b}_i(T - 1)$, $\mathbf{y}_i(T - 1)$, $\mathbf{z}_i(T - 1)$) /* Construct the bundle and update the planning information */
      $t \leftarrow$ Time Reading( ) /* Record the current time */
      $\mathcal{N}_i^t \leftarrow$ Neighbor Getting($t$, $i$, $\{\mathcal{L}_{i1}, \mathcal{L}_{i2}, \ldots, \mathcal{L}_{iN}\}$) /* Calculate the communication availability */
      **for** $i'$ in $\mathcal{N}_i^t$ **do**
        Planning Information Sending($i$, $i'$, $\mathbf{y}_i(T)$, $\mathbf{z}_i(T)$, $\mathbf{s}_i$) /*Send the planning information to each neighbor */
      **end for**
    **end if**
    **if** neighbor $sat_{i'}$ is sending planning information to $sat_i$ **then**
      Planning Information Receiving($i$, $i'$, $\mathbf{y}_{i'}$, $\mathbf{z}_{i'}$, $\mathbf{s}_{i'}$) /*Receive the planning information from the communicating neighbor */
      $t \leftarrow$ Time Reading( ) /* Record the time of current communication */
      $conf_i$, $\mathbf{b}_i(T)$, $\mathbf{y}_i(T)$, $\mathbf{z}_i(T)$, $\mathbf{s}_i \leftarrow$ Consensus Check($t$, $i$, $i'$, $\mathcal{I}$, $\mathcal{J}$, $\{\mathcal{K}_{i1}, \mathcal{K}_{i2}, \ldots, \mathcal{K}_{iM}\}$, $\mathbf{b}_i(T)$, $\mathbf{y}_i(T)$, $\mathbf{z}_i(T)$, $\mathbf{s}_i$, $\mathbf{y}_{i'}$, $\mathbf{z}_{i'}$, $\mathbf{s}_{i'}$) /* Check the consensus between satellite itself and its neighbor and update the planning information */
    **end if**
  **end while**
**end procedure**

---

### 3.2. Bundle Construction

In this phase, each satellite iteratively adds tasks to its bundle until all tasks are added to the bundle or cannot be added due to constraint violation. The construction process is sequentially greedy, each time selecting one task to add to the bundle. The selected task needs to meet two sufficient conditions: (1) $DMG_{ij} > y_{ij}$, i.e., $h_{ij} = 1$; (2) its $DMG_{ij}$ is the largest among all the tasks that meet condition (1). In each iteration, the procedure

of bundle construction for $sat_i$ repeats the following steps until all tasks are added to the bundle or no task can be added because condition (1) is not met:

(1) Calculate the DMG of each task that have not been added to $\mathbf{b}_i$. The DMG of $task_j$ can be calculated as:

$$DMG_{ij} = \max\left(fp_j^{\text{task}}\right) = \max\left(op_j^{\text{task}}e^{-\lambda st_j^{\text{task}}}\right), \forall j \in \mathcal{J}\backslash\mathbf{b}_i \tag{11}$$

where $fp_j^{\text{task}}$ increases with the decrease in $st_j^{\text{task}}$. Therefore, to maximize $fp_j^{\text{task}}$, $st_j^{\text{task}}$ needs to be minimized. Based on Formula (6), $DMG_{ij}$ can be obtained when $task_j$ is assigned to the earliest observation availability mask and the task execution start time is equal to the mask start time, which is expressed as:

$$k_{ij}^{\text{DMG}} = \text{argmin}_k\left(st_{ijk}^{\text{oam}}\right), \quad \forall k \in \mathcal{K}_{ij} \text{ s.t. } f_{ijk} = 1 \tag{12}$$

$$st_{ij}^{\text{DMG}} = st_{ijk_{ij}^{\text{DMG}}}^{\text{oam}}. \tag{13}$$

(2) Select the task that meets two sufficient conditions. The selected task index $j^{\text{add}}$ can be expressed as:

$$j^{\text{add}} = \text{argmax}_j\left(DMG_{ij} \cdot h_{ij}\right), \forall j \in \mathcal{J}\backslash\mathbf{b}_i \text{ s.t. } h_{ij} = 1. \tag{14}$$

(3) Update its bundle $\mathbf{b}_i$, the winning satellite list $\mathbf{z}_i$, and the winning bid list $\mathbf{y}_i$. If $sat_i$ selects $task_{j^{\text{add}}}$, it will enter the task index into its bundle $\mathbf{b}_i$

$$B_i = B_i + 1 \tag{15}$$

$$b_{iB_i} = j^{\text{add}} \tag{16}$$

enter its own index into the winning satellite list $\mathbf{z}_i$

$$z_{ij^{\text{add}}} = i \tag{17}$$

and enter its corresponding DMG into the winning bid list $\mathbf{y}_i$

$$y_{ij^{\text{add}}} = DMG_{ij^{\text{add}}}. \tag{18}$$

(4) Check constraints and delete masks that conflict with the task added to the bundle this time. After a task is successfully added, it is necessary to perform a constraint check on the masks of all tasks outside the bundle. Check whether each mask satisfies the storage, energy and temporary constraints shown in Formulas (3), (4), (6)–(8). If any constraint is not satisfied, change the corresponding mask enable flag $f_{ijk} = 0$. A constraint check makes DMG only calculated for masks with $f_{ijk} = 1$ each time. Because of this, bundle construction can sequentially handle coupled constraints in MSOOP. The pseudo-code of bundle construction is described in Algorithm 2.

---

**Algorithm 2.** Bundle Construction for $sat_i$ at iteration $T$

---

**procedure** Bundle Construction(current satellite index $i$, task index set $\mathcal{J} = \{1, 2, \ldots, M\}$, observation availability mask index sets $\{\mathcal{K}_{i1}, \mathcal{K}_{i2}, \ldots, \mathcal{K}_{iM}\}$, $\mathbf{b}_i(T-1)$, $\mathbf{y}_i(T-1)$, $\mathbf{z}_i(T-1)$)

    Initialize $\mathbf{b}_i(T) = \mathbf{b}_i(T-1)$, $\mathbf{y}_i(T) = \mathbf{y}_i(T-1)$, $\mathbf{z}_i(T) = \mathbf{z}_i(T-1)$, termination flag $TF = 0$

    **while** $TF = 0$ **do** /* Execute the procedure if there is still a task that can be added to the bundle */

        Initialize $j^{\text{add}} = -1$, $DMG_{\text{max}} = 0$

        **for** $j$ in $\mathcal{J}\backslash\mathbf{b}_i$ **do**

            Initialize $DMG_{ij} = 0$, $st_{ij}^{\text{DMG}} = $ planning horizon

---

---

**for** $k$ in $\mathcal{K}_{ij}$ **do**

　**if** $f_{ijk} = 1 \wedge st_{ijk}^{\text{oam}} < st_{ij}^{\text{DMG}}$ **then** /* Find the earliest observation availability mask that meets the constraints */

　　$k_{ij}^{\text{DMG}} = k$

　　$st_{ij}^{\text{DMG}} = st_{ijk_{ij}^{\text{DMG}}}^{\text{oam}}$ /* Record the start time of the earliest observation availability mask found so far */

　　$DMG_{ij} = op_j^{\text{task}} e^{-\lambda st_{ij}^{\text{DMG}}}$ /* Calculate the DMG of the task */

　**end if**

**end for**

$h_{ij} = \mathbb{I}\left(DMG_{ij} > y_{ij}\right)$

**if** $h_{ij} = 1 \wedge DMG_{ij} > DMG_{\max}$ **then** /* Find the task that meets two sufficient conditions */

　$j^{\text{add}} = j$ /* Record the selected task index */

　$DMG_{\max} = DMG_{ij}$ /* Record the maximum DMG found so far */

**end if**

**end for**

**if** $j^{\text{add}} \neq -1$ **then** /* Judge whether there is a task that can be successfully added to the bunde */

　$B_i = B_i + 1$ /* Extend the length of the bundle */

　$b_{iB_i} = j^{\text{add}}$ /* Enter the task into the bundle */

　$y_{ij^{\text{add}}} = DMG_{ij^{\text{add}}}$ /* Enter the DMG of the task into the winning bid list */

　$z_{ij^{\text{add}}} = i$ /* Enter the satellite index into the winning satellite list */

　**for** $j$ in $\mathcal{J} \backslash \mathbf{b}_i$ **do** /* Check constraints and delete masks that conflict with the task added to bundle this time */

　　**for** $k$ in $\mathcal{K}_{ij}$ **do**

　　　**if** $oam_{ijk}$ conflicts with $task_{j^{\text{add}}}$ **then**

　　　　$f_{ijk} = 1$

　　　**else**

　　　　$f_{ijk} = 0$

　　　**end if**

　　**end for**

　**end for**

**else**

　$TF = 1$

**end if**

**end while**

**end procedure**

---

### 3.3. Consensus Check

In MSOOP, the time-varying characteristic of the inter-plane ISLs makes the communication network a dynamic network. However, there are no isolated nodes in the whole constellation, which ensures that ICBBA can eventually converge. The global consensus is the condition of algorithm convergence. A consensus check phase ensures that the final planning scheme must meet the consensus constraint described in Formula (5).

After receiving the planning information from the neighbor $sat_{i'}$ at time $t$, the procedure of consensus check for $sat_i$ includes the following steps:

(1) Check whether the consensus between $sat_i$ and $sat_{i'}$ has been reached. If the consensus has been reached, make the consensus flag $conf_i = 1$; otherwise, change $\mathbf{y}_i$, $\mathbf{z}_i$ based on the action rule shown in Table 1, and make the consensus flag $conf_i = 0$.

**Table 1.** Action rule for $sat_i$ based on communication with $sat_{i'}$ regarding $task_j$.

| $sat_{i'}$ (Sender) Thinks $z_{ij}$ Is | $sat_i$ (Receiver) Thinks $z_{ij}$ Is | Action of $sat_i$ (Default: Leave) |
|---|---|---|
| | $i$ | if $y_{i'j} > y_{ij} \rightarrow$ update |
| $i'$ | $i'$ | update |
| | $i'' \notin \{i, i'\}$ | if $s_{i'i''} > s_{ii''}$ or $y_{i'j} > y_{ij} \rightarrow$ update |
| | none | update |
| | $i$ | leave |
| $i$ | $i'$ | reset |
| | $i'' \notin \{i, i'\}$ | if $s_{i'i''} > s_{ii''} \rightarrow$ reset |
| | none | leave |
| | $i$ | if $s_{i'i''} > s_{ii''}$ and $y_{i'j} > y_{ij} \rightarrow$ update |
| | $i'$ | if $s_{i'i''} > s_{ii''} \rightarrow$ update<br>else $\rightarrow$ reset |
| $i'' \notin \{i, i'\}$ | $i''$ | $s_{i'i''} > s_{ii''} \rightarrow$ update |
| | $i''' \notin \{i, i', i''\}$ | if $s_{i'i''} > s_{ii''}$ and $s_{i'i'''} > s_{ii'''} \rightarrow$ update<br>if $s_{i'i''} > s_{ii''}$ and $y_{i'j} > y_{ij} \rightarrow$ update<br>if $s_{i'i'''} > s_{ii'''}$ and $s_{ii''} > s_{i'i''} \rightarrow$ reset |
| | $i$ | leave |
| none | $i'$ | reset |
| | $i'' \notin \{i, i'\}$ | if $s_{i'i''} > s_{ii''} \rightarrow$ update |
| | none | leave |

There are three possible actions $sat_i$ can take on $task_j$:

- Update: $y_{ij} = y_{i'j}$, $z_{ij} = z_{i'j}$;
- Reset: $y_{ij} = 0$, $z_{ij} = -1$;
- Leave: $y_{ij} = y_{ij}$, $z_{ij} = z_{ij}$.

(2) Update the time stamp $\mathbf{s}_i$. If $conf_i = 1$, end the procedure of this phase; otherwise, continue to the next steps.

$$s_{ii'} = t \tag{19}$$

$$s_{ii''} = \max\{s_{ii''}, s_{i'i''}\}, \forall i'' \in \mathcal{I} \setminus \{i, i'\}. \tag{20}$$

(3) Release the earliest added task for which $z_{ij} \neq i$ and all of the tasks that were added after it. $\overline{n}$ is introduced to record the position of the first released task in $\mathbf{b}_i$.

$$y_{ib_{in}} = 0, \ \forall n > \overline{n} \tag{21}$$

$$z_{ib_{in}} = -1, \ \forall n > \overline{n} \tag{22}$$

$$b_{in} = \varnothing, \ \forall n \geq \overline{n}. \tag{23}$$

$$B_i = \overline{n} - 1. \tag{24}$$

(4) Recover the masks that have been deleted due to conflict with the released tasks. The constraint check in step (4) of bundle construction results in the deletion of the masks which conflict with the tasks already added. Since the tasks mentioned in step (3) have been released from the bundle, the masks which conflict with the released tasks should be recovered. The pseudo-code of consensus check is described in Algorithm 3.

---

**Algorithm 3.** Consensus Check for $sat_i$ at iteration $T$

---

**procedure** Consensus Check (planning information receiving time $t$, current satellite index $i$, neighbor satellite index $i'$, satellite index set $\mathcal{I} = \{1, 2, \ldots, N\}$, task index set $\mathcal{J} = \{1, 2, \ldots, M\}$, observation availability mask index sets $\{\mathcal{K}_{i1}, \mathcal{K}_{i2}, \ldots, \mathcal{K}_{iM}\}$, $\mathbf{b}_i(T)$, $\mathbf{y}_i(T)$, $\mathbf{z}_i(T)$, $\mathbf{s}_i$, $\mathbf{y}_{i'}$, $\mathbf{z}_{i'}$, $\mathbf{s}_{i'}$)

 Initialize $conf_i = 1, \bar{n} = -1$
 **for** $j$ **in** $\mathcal{J}$ **do**
  **if** $y_{ij} \neq y_{i'j} \vee z_{ij} \neq z_{i'j}$ **then** /* Check the consensus between satellite itself and its neighbor */
   $conf_i = 0$
  **end if**
  $y_{ij}, z_{ij} \leftarrow$ Action Rule($y_{ij}, z_{ij}, y_{i'j}, z_{i'j}$) /* Change the winning bid list and the winning satellite list based on the action rule */
 **end for**
 **for** $i''$ **in** $\mathcal{I}$ **do** /* Update the time stamp */
  **if** $i'' = i'$ **then**
   $s_{ii''} = t$
  **end if**
  **if** $i'' \neq i \wedge i'' \neq i'$ **then**
   $s_{ii''} = \max\{s_{ii''}, s_{i'i''}\}$
  **end if**
 **end for**
 **if** $conf_i = 0$ **then**
  **for** $n = 1$ to $B_i$ **do** /* Release the earliest added task for which $z_{ij} \neq i$ and all of the tasks that were added after it */
   **if** $\bar{n} = -1 \wedge z_{ib_{in}} \neq i$ **then** /* Find the earliest added task for which $z_{ij} \neq i$ */
    $\bar{n} = n$
    $b_{in} = \varnothing$ /* Empty the corresponding location of the bundle */
   **end if**
   **if** $\bar{n} \neq -1$ **then**
    $y_{ib_{in}} = 0$ /* Reset the corresponding location of the winning bid list */
    $z_{ib_{in}} = -1$ /* Reset the corresponding location of the winning satellite list */
    $b_{in} = \varnothing$ /* Empty the corresponding location of the bundle */
   **end if**
  **end for**
  $B_i = \bar{n} - 1$ /* Shorten the length of the bundle */
  **for** $j$ **in** $\mathcal{J} \backslash \mathbf{b}_i$ **do** /* Recover the masks that have been deleted due to conflict with the released tasks */
   **for** $k$ **in** $\mathcal{K}_{ij}$ **do**
    **if** $oam_{ijk}$ conflicts with any task in $\mathbf{b}_i$ **then**
     $f_{ijk} = 1$
    **else**
     $f_{ijk} = 0$
    **end if**
   **end for**
  **end for**
 **end if**
**end procedure**

---

## 4. Computational Experiments

In this section, we present experimental studies on ICBBA implemented in MSOOP. AGI's STK is used to simulate the experimental scenes and calculate the observation availability masks and communication availability masks.

Some details about the experimental instances and environment are as follows:

- Target setting: the targets are divided into two types, global targets and regional targets. The global targets are generated by a random uniform distribution throughout the world, 60° S–60° N. The regional targets are generated by a random uniform distribution in the area, 3° N–53° N and 73° E–133° E. The position of each target is

defined by latitude and longitude. For different types of targets, the number of targets includes 500, 1000, and 1500, respectively.

- Satellite constellation setting: Walker-δ (30/3/1, 600 km, 60°), Walker-δ (60/3/1, 600 km, 60°), and Walker-δ (90/3/1, 600 km, 60°) are used.
- Parameter setting: the planning horizon is set to 1.5 h, from 04:00:00 to 05:30:00 on 30 July 2022. The original profit $op_j^{\text{task}}$ and the estimated consumed storage $sto_j^{\text{task}}$ of each task are generated by a random uniform distribution on $[50, 100]$. The maximum available storage $sto_i^{\text{sat}}$ is set to 750, which allows that about 10 tasks can be completed by each satellite in the whole planning horizon. The time-discounted parameter $\lambda$ in Formula (1) is set to $10^{-5}$, which means that the final profit $fp_j^{\text{task}}$ of the task completed at the last moment of the planning horizon is about 95% of the original profit $op_j^{\text{task}}$.
- Experimental environment: The proposed algorithm was coded in C++ and compiled on an Intel Core i9-11900K processor (3.5 GHz and 64 GB RAM).

To reflect the performance of ICBBA under different constraints, experiments are carried out on global targets and regional targets, respectively. For global targets, due to the scattered distribution of targets, the conflicts between different tasks about observation availability masks are not prominent. The storage constraint and the energy constraint become dominant constraints. The DMG calculation can be flexibly adjusted according to the characteristics of different scenes. The most basic calculation that uses task profit to calculate DMG is described in Formula (11), which is denoted as DMG(P). According to the above characteristics of global targets, the estimated consumed storage $sto_j^{\text{task}}$ is introduced into the DMG calculation, which is expressed as:

$$DMG_{ij} = \max\left(\frac{fp_j^{\text{task}}}{sto_j^{\text{task}}}\right), \forall j \in \mathcal{J} \backslash \mathbf{b}_i. \tag{25}$$

This DMG calculation is denoted as DMG(P/S). DMG(P/S) focuses on those tasks with higher profit per unit storage, which can be seen as the balance between profit and storage. The comparison results of DMG(P) and DMG(P/S) on global targets are shown in Table 2.

**Table 2.** The comparison results of DMG(P) and DMG(P/S) on global targets. (The highest value of task completion rate and total profit in each instance are marked in bold.)

| Target Type | Target Number | Constellation (*N/P/F*) | Available Target Number | Communication Number | DMG(P) | | | DMG(P/S) | | |
|---|---|---|---|---|---|---|---|---|---|---|
| | | | | | Completion Number | Completion Rate (%) | Total Profit | Completion Number | Completion Rate (%) | Total Profit |
| Global targets | 500 | 30/3/1 | 282 | 3552 | 261 | 92.55 | 20,134.6 | 263 | **93.26** | **20,201.9** |
| | | 60/3/1 | 299 | 12,434 | 297 | **99.33** | **22,233.5** | 297 | **99.33** | 22,233.4 |
| | | 90/3/1 | 314 | 24,786 | 312 | 99.36 | 23,313.3 | 313 | **99.68** | **23,384.9** |
| | 1000 | 30/3/1 | 556 | 3241 | 289 | 51.98 | 24,863.2 | 314 | **56.47** | **25,553.5** |
| | | 60/3/1 | 607 | 19,211 | 574 | **94.56** | **43,360.4** | 569 | 93.74 | 42,934.3 |
| | | 90/3/1 | 629 | 37,821 | 621 | **98.73** | **46,406.8** | 621 | **98.73** | 46,361.9 |
| | 1500 | 30/3/1 | 845 | 3728 | 289 | 34.20 | 26,117.8 | 339 | **40.12** | **28,147.6** |
| | | 60/3/1 | 882 | 16,702 | 582 | 65.99 | 48,043.5 | 609 | **69.05** | **48,573** |
| | | 90/3/1 | 938 | 47,181 | 866 | **92.32** | **65,918.2** | 864 | 92.11 | 65,601 |

For regional targets, due to the dense distribution of targets, there are more conflicts reflected in the temporary constraints. In order to further highlight the dominance of the temporary constraints, the storage constraint and the energy constraint are not considered. To describe the impact of the execution of one task on other tasks, a new concept "Loss" is introduced into the DMG calculation, which is expressed as:

$$DMG_{ij} = \max\left(fp_j^{\text{task}} - l_j^{\text{task}}\right), \forall j \in \mathcal{J} \backslash \mathbf{b}_i \tag{26}$$

where $l_j^{\text{task}}$ is the expected profit sum of those masks which conflict with the final assigned mask in the temporary constraints. This DMG calculation is denoted as DMG(P-L). Due to the temporary constraints, the execution of one task may lead to the loss caused by the failure of other related tasks. DMG(P-L) focuses on not only the profit of the task itself but also the loss mentioned above. The comparison results of DMG(P) and DMG(P-L) on regional targets are shown in Table 3.

**Table 3.** The comparison results of DMG(P) and DMG(P-L) on regional targets. (The highest value of task completion rate and total profit in each instance are marked in bold.)

| Target Type | Target Number | Constellation (*N/P/F*) | Available Target Number | Communication Number | DMG(P) | | | DMG(P-L) | | |
|---|---|---|---|---|---|---|---|---|---|---|
| | | | | | Completion Number | Completion Rate (%) | Total Profit | Completion Number | Completion Rate (%) | Total Profit |
| Regional targets | 500 | 30/3/1 | 256 | 4321 | 222 | 86.72 | 16,472.5 | 224 | **87.50** | **16,488.1** |
| | | 60/3/1 | 274 | 15,457 | 262 | 95.62 | 19,161.4 | 272 | **99.27** | **19,666.8** |
| | | 90/3/1 | 282 | 33,502 | 277 | 98.23 | 20,861 | 281 | **99.65** | **20,992.1** |
| | 1000 | 30/3/1 | 546 | 5641 | 370 | 67.77 | 28,473.4 | 374 | **68.50** | **28,548.2** |
| | | 60/3/1 | 580 | 22,816 | 510 | 87.93 | 38,045.9 | 524 | **90.34** | **38,773** |
| | | 90/3/1 | 586 | 46,540 | 558 | 95.22 | 41,911.1 | 577 | **98.46** | **42,943.7** |
| | 1500 | 30/3/1 | 836 | 5627 | 426 | 50.96 | 34,464.7 | 454 | **54.31** | **35,541.6** |
| | | 60/3/1 | 876 | 24,869 | 680 | 77.63 | 52,207.9 | 690 | **78.77** | **52,341.7** |
| | | 90/3/1 | 864 | 55,574 | 770 | 89.12 | 57,981.3 | 811 | **93.87** | **60,182.9** |

Take "global targets-1500-90/3/1" and "regional targets-1500-90/3/1" as examples; the number of repeated claims for the whole constellation after bundle construction at each iteration is shown in Figure 3a,b respectively. The reduction in the number of repeated claims proves that the consensus check phase plays an important role in the convergence of ICBBA.

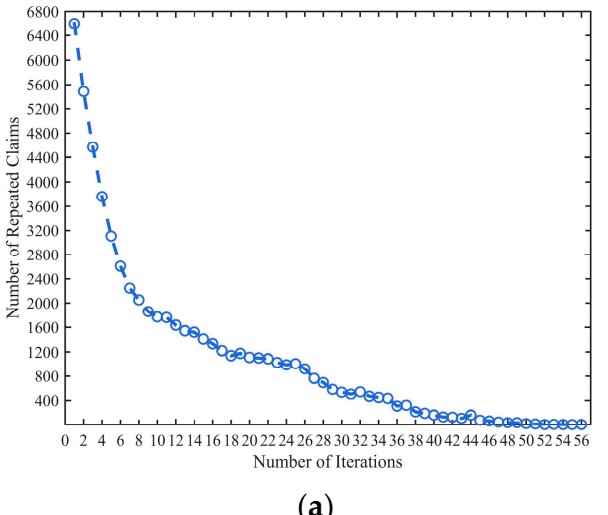

(**a**)

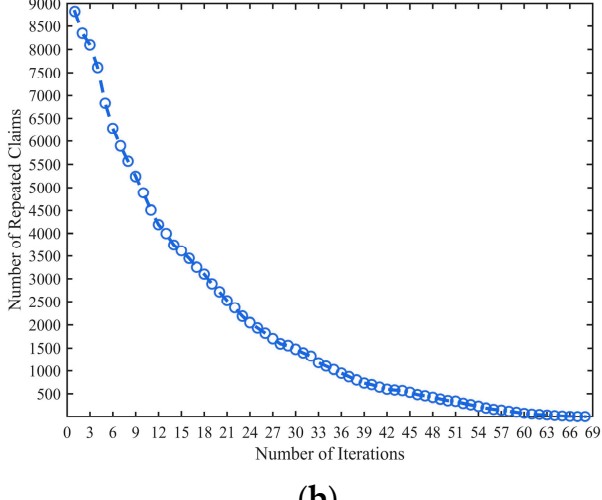

(**b**)

**Figure 3.** The number of repeated claims for the whole constellation after bundle construction varies according to iteration times: (**a**) Global targets-1500-90/3/1; (**b**) Regional targets-1500-90/3/1.

In order to test the responsiveness of ICBBA to dynamically arrived tasks, taking "global targets-1500-90/3/1" and "regional targets-1500-90/3/1" as examples, 100 tasks are set to arrive at the 30th iteration of the algorithm running process. The results of DMG(P) with tasks arriving dynamically are shown in Table 4. The number of repeated claims for the whole constellation after bundle construction at each iteration is shown in Figure 4a,b, respectively, which records the process of re-convergence in the face of dynamically arrived tasks during the running process.

**Table 4.** The results of DMG(P) on global targets and regional targets with tasks arriving dynamically.

| Target Type | Target Number | Constellation (*N/P/F*) | Available Target Number | Communication Number | DMG(P) | | |
|---|---|---|---|---|---|---|---|
| | | | | | Completion Number | Completion Rate (%) | Total Profit |
| Global targets | 1500 | 90/3/1 | 938 | 47,181 | 866 | 92.32 | 65,918.2 |
| | 1500 + 100 | 90/3/1 | 994 | 49,785 | 870 | 87.53 | 67,426.7 |
| Regional targets | 1500 | 90/3/1 | 864 | 55,574 | 770 | 89.12 | 57,981.3 |
| | 1500 + 100 | 90/3/1 | 916 | 58,520 | 813 | 88.76 | 61,280.5 |

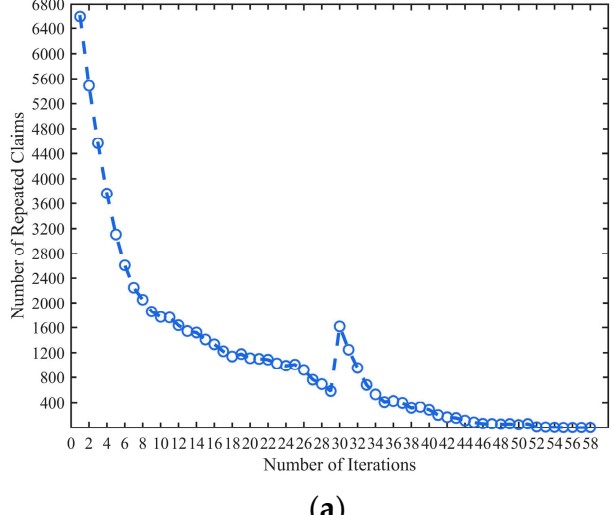

(**a**)

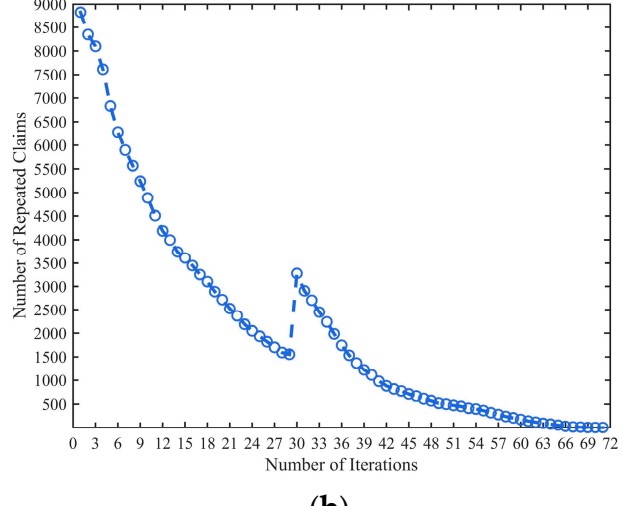

(**b**)

**Figure 4.** The number of repeated claims for the whole constellation after bundle construction varies according to iteration times, with 100 tasks arriving dynamically at the 30th iteration: (**a**) Global targets-1500-90/3/1; (**b**) Regional targets-1500-90/3/1.

From the experimental results, the following observations can be made:

(1) Communication cost analysis: From Tables 2 and 3, we can see that for both global and regional targets, the communication number increases with the number of satellites *N* and the number of tasks *M*. However, there are still a few exceptions, such as "global targets-500-30/3/1" and "global targets-1000-30/3/1". The target number increases from 500 to 1000 and the available target number increases from 282 to 556, while the communication number decreases from 3552 to 3241. Therefore, this positive proportional relationship is not simply linear or exponential. Compared with other decentralized algorithms, such as Contract Network Protocol (CNP) [18], the communication number of ICBBA is significantly lower than "$N \cdot M$". In addition, by comparing each corresponding instance in regional and global targets, the communication number for regional targets is always higher than that for global targets. Because there are more conflicts between tasks for regional targets, this coupling needs more communication to resolve.

(2) Optimization performance analysis: It has been proved that CBBA is equivalent to the centralized sequential greedy algorithm in terms of optimization performance [24], and the rules for greedy can be controlled by designing the DMG calculation function. For global targets, DMG(P/S) has a better performance in MSOOP than DMG(P); similarly, for regional targets, DMG(P-L) has a better performance than DMG(P). In comparison, DMG(P-L) (all better than DMG(P)) has a more significant effect on regional targets than DMG(P/S) (five out of nine better than DMG(P)) on global targets. For global targets, when the storage constraint is not tight, the completion rate is close to 100%; when the storage constraint is tight, the storage can also be effectively used to maximize the completion of tasks.

(3) Convergence performance analysis: From Table 4, it can be seen that ICBBA can effectively handle dynamically arrived tasks and improve the completion rate and the total profit. For "global targets-1500-90/3/1" in Figure 3a, ICBBA iterates 56 times to achieve the

global consensus; for "regional targets-1500-90/3/1" in Figure 3b, ICBBA iterates 68 times to achieve the global consensus. The number of repeated claims decreases with the number of iterations in the whole constellation. The decline speed is faster in the early stage. From Figure 4a,b, it can be seen that when 100 tasks arrive dynamically at the 30th iteration, the number of repeated claims rises sharply and then continues to decline until convergence at the 58th and 71st iteration, respectively.

## 5. Conclusions

In this paper, we investigate the multi-satellite on-board observation planning (MSOOP) with the case study of LEO satellite constellations. MSOOP, as a variant of multi-agent task allocation problem (MATAP), should complete the task allocation in a fully cooperative mode to maximize the profits of the whole constellation.

We give a detailed description of MSOOP to discover its unique characteristics compared with the task allocation model in the original work of CBBA. In this case, not only the commonalities of MSOOP but also some coupled constraints should be taken into consideration, such as the storage constraint, the energy constraint, the consensus constraint and the special temporary constraints, which increases the difficulty of planning.

Adapted to the characteristics of the problem, the improved Consensus-Based Bundle Algorithm (ICBBA) is exploited. ICBBA includes two main phases: bundle construction and consensus check. Each satellite runs the above two phases iteratively in parallel until the global consensus is reached and receives its own final planning scheme to perform observation. A constraint check and a mask recovery are introduced into bundle construction and consensus check to handle the coupled constraints. In addition, the DMG calculation is adjusted to adapt to the characteristics of different scenes.

To validate the effectiveness of the proposed algorithm, we carry out experiments on series instances with different targets and constellations. From the experimental results, we can see that, the communication number increases with the number of satellites and the number of tasks, but it is not simply linear or exponential. ICBBA greatly saves the communication cost compared with CNP. The optimization performance of ICBBA can be controlled by designing the DMG calculation function. Using DMG(P/S) on global targets and DMG(P-L) on regional targets can lead to a better solution. The number of repeated claims decreases with the number of iterations in the whole constellation, which proves that the consensus check phase plays an important role in the convergence of ICBBA. In addition, ICBBA can effectively handle dynamically arrived tasks and rapidly achieve re-convergence.

**Author Contributions:** Conceptualization, B.S. and Q.Y.; methodology, B.S. and Q.Y.; software, B.S.; validation, Y.Z. and S.X.; data curation, Y.Z. and S.X.; writing—original draft preparation, B.S. and Q.Y.; writing—review and editing, Y.C. (Yuning Chen) and Y.C. (Yingwu Chen); visualization, Y.Z. and S.X.; supervision, Y.C. (Yuning Chen) and Y.C. (Yingwu Chen); project administration, Y.C. (Yuning Chen) and Y.C. (Yingwu Chen); funding acquisition, Y.C. (Yuning Chen) and Y.C. (Yingwu Chen) All authors have read and agreed to the published version of the manuscript.

**Funding:** This work was supported by the National Natural Science Foundation of China (Grant numbers: 72001212, 72201272, 71901213).

**Data Availability Statement:** The instances for MSOOP used in this paper can be downloaded at https://github.com/haohaoxuexifalunwen/ICBBA (accessed on 3 January 2023).

**Conflicts of Interest:** The authors declare no conflict of interest.

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
