# Peer review of "On-Board Decentralized Observation Planning for LEO Satellite Constellations"

_algorithms, doi:10.3390/a16020114_

Round 1

Reviewer 1 Report

This paper deals with the planning of missions for LEO satellites constellations. Authors propose an improvement of the Consensus-Based Bundle Algorithm (ICBBA) relative to coupled constraints and fitness function.

The presentation is clear and the propose algorithms composing ICCBA are well detailed. The obtained results are encouraging still, several points are missing the submission or can help the submission to gain in clarity.

First, the state of the art is very brief. Authors does not really justify their choices. I completely agree with their arguments concerning decentralization Vs centralization. Still, several decentralized approaches exist, why CBBA?

The problem presentation is sound and clear.

The algorithms presentation introduces a lot of abbreviations which make this section difficult to follow. It can really help the paper to add comment to the proposed algorithm in addition to the descriptions given in the text. A small example that illustrates each step of the algorithm can also be very useful.

Concerning the proposed algorithm, a question that remains on mind, how tasks are distributed on the constellations, and is that possible that some conflicts are not detected among the satellites? Is there any guarantee that no task will be planned over two satellites?

Another point concerns the fact that a zone on earth is visible to the satellite over a period of time while imaging this zone is very fast. This point increases the complexity of the problem as it increases the possibilities for choosing the best moment to plan each image in the visibility period.  It is not clear in the paper how this choice is done?  Is everything computed before starting the bundle construction process?

Another concern is about the dynamics, on-board planning is indeed a powerful technic to make missions planning more flexible and easier to adapt when new urgent observations are to be done. The paper does not discuss the integration at real-time of new environment conditions.

Concerning the results section, the aim of the paper is to improve a state-of-the-art method. Still, the results do not underline this improvement as no comparison with CBBA is proposed. Comparison with other centralized methods or not on-board algorithm are also appreciable to underline the quality of the proposed approach.

Author Response

Dear reviewer,

Sorry to submit the response so late. Please see the attachment.

Sincerely,

The authors

Reviewer 2 Report

The manuscript is overall well organized, well written, and the results are encouraging. However, the following questions should be addressed before the reviewer can recommend the manuscript for acceptance.

1. It is unclear whether this work is about an application of the ICBBA algorithm or it introduced a new algorithm called ICBBA. Please rephrase wherever necessary to more explicitly clarify your contributions.

2. According to the authors, ICBBA is an improved version of CBBA, why is ICBBA not directly or thoroughly compared with CBBA in the tests? Why is ICBBA compared with another approach, namely, CNP? This is not logically convincing. Please explain and make revisions and further studies/verifications if necessary.

3. When you first mentioned Tables 2 & 3 and Figure 3, there is no discussion at all. Instead, a very brief discussion is given from line 364 to 376. It will be more helpful to provide more detailed/extensive discussion to support your major conclusions.

Author Response

(The authors gave the same response as above.)

Round 2

Reviewer 1 Report

The answers provided by the authors are sound and clear and improve the paper. Two points :

1. Concerning the state of the art, I don't think that CNP is the only distributed algorithm. Indeed there are many other algorithms used for flexible job shop scheduling which is similar to planning LEO satellites missions. Still, the presented state of art is more adequate then the first presented one.

2. Concerning the comparison, I understand your answer. Still, you mainly compare ICCBA using different computation of DMG on measure that can also be obtained using CBBA or a greedy algorithm. I agree that greedy algorithms are not adequate for decentralization issue). Still, it is difficult to claim that you improve a method if you do not show that your results improve it.

The experimentation concerning the dynamic arrival of tasks is appreciated.

Added new paragraphs require minor spell check.

Author Response

Dear reviewer,

Thank you for your review of this paper. Please see the attachment.

Yours sincerely,

All authors

Reviewer 2 Report

The revised paper is much improved and the authors properly addressed all my questions. I thus recommend the paper for acceptance.

Author Response

Dear reviewer,

Thank you for your review of this paper. Wish you all the best!

Yours sincerely,

All authors